# Fast and multiplexed superresolution imaging with DNA-PAINT-ERS

Fehmi Civitci[1,5], Julia Shangguan [2,3,5], Ting Zheng[1,5], Kai Tao[2,3,5], Matthew Rames[1,3], John Kenison[1], Ying Zhang[2,3], Lei Wu[2,4], Carey Phelps[2,3], Sadik Esener [1,3] & Xiaolin Nan [1,2,3✉]

DNA points accumulation for imaging in nanoscale topography (DNA-PAINT) facilitates multiplexing in superresolution microscopy but is practically limited by slow imaging speed. To address this issue, we propose the additions of ethylene carbonate (EC) to the imaging buffer, sequence repeats to the docking strand, and a spacer between the docking strand and the affinity agent. Collectively termed DNA-PAINT-ERS (E = EC, R = Repeating sequence, and S = Spacer), these strategies can be easily integrated into current DNA-PAINT workflows for both accelerated imaging speed and improved image quality through optimized DNA hybridization kinetics and efficiency. We demonstrate the general applicability of DNA-PAINT-ERS for fast, multiplexed superresolution imaging using previously validated oligonucleotide constructs with slight modifications.

[1] Knight Cancer Early Detection Advanced Research Center, Oregon Health and Science University, 2720 S. Moody Ave., Portland, OR 97201, USA. [2] Center for Spatial Systems Biomedicine, Oregon Health and Science University, 2730 S. Moody Ave., Portland, OR 97201, USA. [3] Department of Biomedical Engineering, Oregon Health and Science University, 3303 S. Bond Ave., Portland, OR 97239, USA. [4] Department of Oral Maxillofacial-Head Neck Oncology, School and Hospital of Stomatology, Wuhan University, 237 Luoyu Rd., Wuhan 430079 Hubei, China. [5] These authors contributed equally: Fehmi Civitci, Julia Shangguan, Ting Zheng, Kai Tao. ✉email: nan@ohsu.edu

For single-molecule localization microscopy (SMLM) techniques such as DNA-PAINT[1–5], both the imaging speed and image quality depend on the localization kinetics[6,7]. A fast onset of localization events allows efficient sampling of the target to develop a super-resolved image quickly. At the same time, the events should also vanish quickly—ideally matching the speed of image acquisition—to mitigate spatially overlapping localizations which would otherwise degrade resolution and image quality. In DNA-PAINT, the localizations arise from reversible hybridizations between a docking strand (DS) oligo immobilized on the target and a complementary, fluorophore-conjugated imager strand (IS) oligo diffusing in solution (Fig. 1, first panel). This imaging scheme simplifies multiplexed SMLM by eliminating the need for photo-switchable fluorophores and allows multiple targets to be DNA-barcoded and imaged sequentially[4]. To date, over 15 DS-IS pairs have been validated (out of >50 designed) for multiplexed DNA-PAINT imaging of cellular structures[8].

However, a practical hurdle to using DNA-PAINT for multiplexed SMLM is the slow imaging speed, with each target taking tens of minutes to hours to complete[4]. This is primarily due to relatively slow localization kinetics. On average, the duration of localization events ($\tau_{on}$) in DNA-PAINT is on the order of seconds (s)[9], in sharp contrast to other SMLM ($\tau_{on}$ on the scale of 0.01–0.1 s)[6]; the longer lasting events in turn demand a slower onset of events to reduce overlapping localizations. Besides prolonging data acquisition, the slow kinetics can also degrade image quality, necessitating special imaging schemes such as flat-top structured illumination[9]. DNA-PAINT via Föster resonance energy transfer (FRET-PAINT) affords fast acquisition but suffers a low photon yield[10,11] and, for unknown reasons, a rapid loss of localization events as soon as imaging begins (ref. [12] and unpublished data). More recently, a DS-IS pair referred to as PS3 was found to exhibit fast on-off kinetics to speed up DNA-PAINT by about an order of magnitude[13]. At present, it remains unclear how many orthogonal DS-IS pairs like PS3 exist, and other generally applicable methods are necessary for fast and multiplexed DNA-PAINT in biological applications.

To address these issues, we aim to devise strategies for expediting DNA-PAINT that: (a) are compatible with a large panel of DS-IS pairs such as those previously validated for DNA-PAINT; (b) can be easily integrated into current DNA-PAINT workflows; and thus (c) are readily adopted for multiplexed superresolution imaging. Our rationale is that the key step toward fast and high-quality DNA-PAINT imaging is to speed up DS-IS unbinding, ideally without slowing down the binding. This is achieved by including a small molecule, ethylene carbonate (EC), in the same imaging buffer (buffer C) as used in current DNA-PAINT experiments (Fig. 1, second panel). Next, we increase DS-IS binding via two simple strategies: increasing the copy number of the complementary sequence on the DS by using sequence repeats (Fig. 1, third panel), and increasing the accessibility of the DS by inserting a small spacer between the DS and the affinity

agent to reduce the steric hindrance (Fig. 1, fourth panel). These strategies, collectively termed DNA-PAINT-ERS (where E = EC, R = Repeating sequence, and S = Spacer), allow us to complete multiplexed DNA-PAINT imaging in merely 2–5 minutes (min) for each target. In addition, we show that DNA-PAINT-ERS significantly improve the quality of the resulting images over current DNA-PAINT, likely also a result of better localization kinetics and accessibility of the DS.

## Results

**Accelerated DS-IS unbinding by ethylene carbonate.** EC is a water-soluble, aprotic solvent previously identified as a low-toxicity substitute for formamide in fluorescence in situ hybridization[14]. A 1:1 mixture of EC and water (v/v) as the solvent was shown to dramatically speed up hybridization between target DNA and oligonucleotide probes, possibly by improving the solubility of the hydrophobic core of the bases. Surprisingly, we found that EC actually accelerated the dehybridization of DS-IS with little impact on the reverse process when added to a DNA-PAINT imaging buffer (buffer C[4]; see Methods) at concentrations as low as 5% (v/v). We first noticed a significant decrease in the instantaneous number of IS probes bound to the sample upon addition of EC (Fig. 2a, left panel, top row; and Supplementary Movie 1). Of note, all images shown in the left panel of Fig. 2a were acquired from the same field of view (FOV) on the same sample via careful buffer exchange between imaging cycles without shifting the FOV. To understand how EC affects the kinetics of DS-IS hybridization, we measured the $\tau_{on}$ for individual localization events using a pair of oligos referred to as DS1-IS1. The DS1-IS1 pair has a 10-base complementary sequence derived from a previously published DNA-PAINT construct 'P1' (9 bp; see Supplementary Table 1), with an A appended at the 3'-end of the DS. On average, the $\tau_{on}$ for DS1-IS1 showed a steady decrease at increasing EC concentrations, from >2 s in the absence of EC to ~0.4 s and ~0.2 s at 10% and 15% EC, respectively (Fig. 2a, middle panel; Supplementary Fig. 1). By contrast, the binding rate between DS1 and IS1, measured as the sorted number of localization events per unit time (frame), remained largely the same (Fig. 2a, right panel). Here sorting refers to the process of combining events that likely arise from the same molecules (see Methods). Thus, by using imaging buffers containing 10–15% EC, the rate of DS-IS unbinding could be accelerated by 5–10-fold without reducing the rate of binding.

The accelerated unbinding and unaffected binding rates between the DS and the IS can also be seen by examining the difference images between successive image frames. In particular, binding events could be detected by subtracting the previous image frame from the current (Fig. 2a, left panel, middle row) and unbinding by doing the opposite (Fig. 2a, left panel, bottom row). At each EC concentration, the net number of binding events per frame was similar to that of the unbinding, which reflects the fact

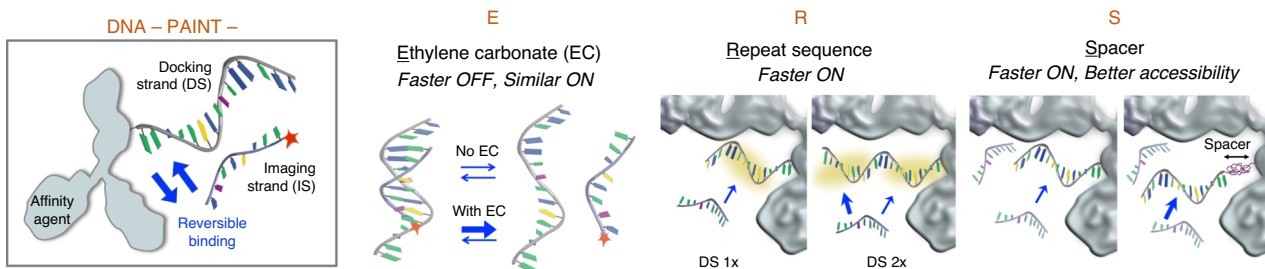

**Fig. 1 Schematics of DNA-PAINT-ERS.** First panel illustrates the standard DNA-PAINT process. Second to fourth panels depict the effect of ethylene carbonate (E), repeating sequence (R), and spacer (S), respectively. In the latter two cases, the DNA oligos and the antibody are drawn roughly to scale.

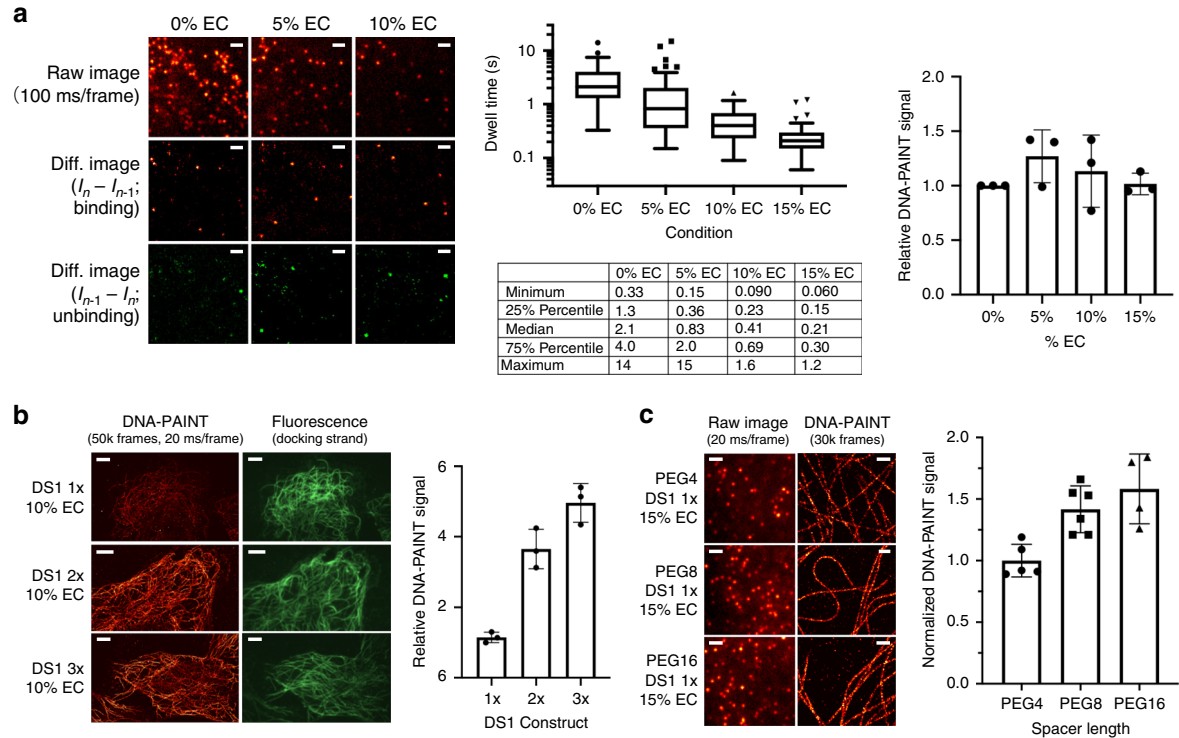

**Fig. 2 Accelerating DNA-PAINT with E, R, and S. a** EC accelerates DS-IS unbinding. First panel, single-molecule images of the same sample region at different EC concentrations (top) and the corresponding difference images for binding (the current frame minus the preceding frame; middle) and the reverse (unbinding; bottom). Middle panel shows dwell time ($\tau_{on}$; plot at the top and key statistics at the bottom) at different EC%. Right panel shows the sorted number of localizations normalized to that at 0% EC. Images were acquired on U2OS cells labeled for microtubules using 0.25 nM IS1-CF660R at a low power density (~100 W cm$^{-2}$) to minimize photobleaching. Three regions from two independent samples were analyzed at each EC concentration. **b** Sequence repeats speed up DS-IS hybridization. Left, exemplary images of microtubules in U2OS cells using DS1 with 1x (top), 2x (middle), and 3x (bottom) repeats, showing reconstructed DNA-PAINT images (left column) and epi-fluorescence images of Cy3-DS1 (right column). Right, DNA-PAINT signal normalized to the corresponding DS signal, at 1.1 ± 0.1, 3.7 ± 0.6, and 5.0 ± 0.6 (mean ± SD) for DS1 1x, 2x, and 3x, respectively. DNA-PAINT images were acquired in ~17 min (50,000 frames at 20 ms per frame) using 0.2 nM IS1-CF660R and 10% EC. Results from three experiments were analyzed. **c** Spacer between DS and antibody improves IS-DS binding. Left panel, example single-molecule images (left) and reconstructed DNA-PAINT images (right) of microtubules in U2OS cells using DS1-antibody conjugates with a PEG4 (top), a PEG8 (middle), or a PEG16 (bottom) spacer. Images were acquired in 10 min (30,000 frames at 20 ms per frame) using 2 nM IS1-CF660R and 15% EC. Right panel, total DNA-PAINT signals normalized to that using DS1-PEG4-antibody, showing 1.0 ± 0.1, 1.4 ± 0.2, and 1.6 ± 0.3 (mean ± SD) for PEG4, PEG8, and PEG16, respectively. Four to six regions of interest from two independent experiments were sampled for each spacer length. All error bars are SD. Scale bars are 2 μm in (**a**), 5 μm in (**b**), 2 μm in (**c**, left panel, left column), and 500 nm in (**c**, left panel, right column). Source data underlying Fig. 2a–c are provided as a Source Data file.

that the binding and unbinding processes are at equilibrium during DNA-PAINT imaging. By comparing the difference images for binding (Fig. 2a, left panel, middle row) at different EC concentrations, we observed that the numbers of binding events per frame were similar in the full range of EC concentrations tested (0–15%). This confirms that the binding rate between DS1 and IS1 stayed largely unaffected by EC; here, the binding rate is the net number of localizations per unit time normalized to the concentrations of free IS1 in the buffer (kept constant at all EC concentrations) and free DS1 (essentially the same as total DS1 thus also approximately constant). Based on similar analyses, the unbinding rate is inversely proportional to the concentration of DS1-IS1 duplexes or the density of IS1 bound to the sample. Thus, the 5–10-fold reduction in the density of IS on the sample at 10–15% EC (Fig. 2a) corresponds to a 5–10-fold increase in the unbinding rate, or equivalently, a 5–10-fold decrease in the $\tau_{on}$.

**Accelerated DS-IS binding by tandem sequence repeats and spacer.** The accelerated unbinding of DS-IS by EC made it now practical to use strategies to accelerate DS-IS hybridization without concerns of spatially overlapping localizations. One

approach to achieving this is to increase the IS concentration, but this is typically limited to below 3–4 nM due to increased background signal from the diffusing IS. In addition, the binding rate could be significantly increased if there are multiple copies of the complementary (docking) sequences on the DS (Fig. 1, third panel). We therefore constructed new DS1 oligos with 2 or 3 tandem repeats of the docking sequence (referred to as DS1-2x and DS1-3x, respectively) and tested their performance for DNA-PAINT. Indeed, both the 2x and 3x constructs showed dramatically increased binding of IS, resulting in much more continuous structures of the target (microtubules in this case) than using the original 1x construct under the same imaging conditions (Fig. 2b, left). All the DS constructs had a fluorophore (Cy3 or FAM) attached at a 1:1 stoichiometric ratio, allowing us to normalize the DNA-PAINT signal (measured as the total number of sorted localization events) to the total DS signal (measured as the total signal from the fluorophore conjugated to the DS) from the same FOV. Using this normalization approach, we estimated that the use of 2x and 3x DS1 yielded ~3.5x and ~5x faster binding rates, respectively, compared with DS-1x (Fig. 2b, right), demonstrating the effectiveness of using repeating sequences to accelerate the binding between the DS and the IS.

The fact that using 2x and 3x DS constructs increased the binding rates by more than 2 and 3 times, respectively, led us to hypothesize that docking sites on the DS farther away from the affinity agent (a secondary antibody in this case) may be more efficiently probed by the IS, potentially due to reduced steric hindrance (Fig. 1, third and fourth panels). We therefore asked whether inserting a spacer between the DS and the antibody would serve the same purpose. For the antibody-DS conjugates tested thus far, we had used a short spacer comprising a 4-unit polyethyleneglycol (PEG4) between the antibody and the DS. The spacer is part of a reagent used to prepare the conjugates and can be conveniently replaced with a different moiety (see Methods). When we extended the spacer on DS1-1x to PEG8 or PEG16, we observed a clear acceleration similar to that observed using DS1-2x or -3x, albeit to a lesser extent. The microtubule structure in the reconstructed DNA-PAINT images appeared much more continuous from a 10 min acquisition when using DS1-1x with the longer spacers (Fig. 2c, left). After normalizing to the total DS1 signal, we found the use of PEG8 and PEG16 spacers to increase the binding rate by ~40% and ~60%, respectively (Fig. 2c, right). This result confirms that the effect of using DS1-2x or -3x was indeed in part due to the alleviated steric hindrance on the 2nd and 3rd repeats of the docking sequence. We note that the spacer strategy was previously used to expedite hybridization to surface-anchored DNA[15], a situation similar to that in DNA-PAINT where the DS is typically immobilized on the target.

We further verified the effects of R (repeating sequence) and S (spacer) on DNA-PAINT imaging using another DS-IS construct. The new DS, which we designated DS2, was derived from a previously validated DNA-PAINT construct 'P2' by adding an extra T to the 3′-end (see Supplementary Table 1). Instead of comparing DS2-1x with DS2-2x or DS2-PEG4 with DS2-PEG8/16 as in the case of DS1, here we chose to use three IS oligos that recognize different parts of the same DS2-2x-PEG4 construct to vary the number of sequence repeats or the spacer length (Supplementary Fig. 2a–c, left panels). The first IS oligo, IS2-A, recognizes two docking sequences on the DS-2x with the first one separated from the antibody by two bases (plus the PEG4), rendering the DS2-2x a true 2x construct. The second and the third IS oligos, IS2-B and IS2-C, have only one binding site on the DS2-2x located two and four bases further downstream of the 1st binding site for IS2-A. Of the three, IS2-B was used as a normalization standard (though not perfect) because it has half the binding sites as IS2-A and a shorter spacer than IS2-C. This experimental scheme allowed us to image the same sample FOV using the three IS oligos in a sequential manner, so we could directly compare how the different IS performed against the same DS. As shown in Supplementary Movie 2, binding rates of IS2-A and IS2-C were both much faster than that of IS2-B, with IS2-A clearly being the fastest. The differences were also evident in the difference and the reconstructed images (Supplementary Fig. 2a–c, right panels). Quantitation of the raw and reconstructed images revealed that the binding rate of IS2-A to DS2-2x-PEG4 was ~3x that of IS2-B, and that IS2-C was 30–40% faster than IS2-B despite the seemingly small increase in spacer length by only two bases (Supplementary Fig. 2d, e). These results are comparable to our observations on DS1-IS1 (Fig. 2b, c) and suggest that accelerated DS-IS binding by R and S are not specific to a particular DNA-PAINT construct. In addition, the comparisons imply that a segment of extra (non-complementary) nucleotides inserted between the affinity agent and the DS (e.g., the six bases on the 5′-end of DS2-2x when using IS2-C) could be used as spacers in place of PEG. In fact, in the DS-2x constructs the first docking sequence (located closer to the affinity agent) essentially served as a spacer for the second. For the remainder of this work, we continued using PEG oligomers as the spacer.

**Combining E, R, and S for fast and multiplexed DNA-PAINT.**
By combining the three components (EC, Repeating sequence, and Spacer), DNA-PAINT-ERS obtains fully developed images of cellular structures in a matter of minutes. For example, by using the DS1-2x construct attached to an anti-mouse secondary antibody via a PEG16 linker (DS1-2x-PEG16), we were able to image microtubules in detergent-extracted U2OS cells in merely 150 s (Fig. 3a). DNA-PAINT-ERS imaging of non-extracted cells typically warrants a longer acquisition due to slower probe diffusion, which could both decrease binding rate and cause higher background, but the imaging was still routinely completed within 200–300 s (Supplementary Fig. 3). In either case, the significantly accelerated DS-IS localization kinetics yielded clean and bright single-molecule images (Supplementary Movies 3 and 4) to afford a lateral localization precision better than 10 nm, corresponding to a spatial resolution of ~23 nm or better, consistent with the apparent width of microtubules measured in the resulting images (Fig. 3a, right panels).

DNA-PAINT-ERS is also readily applicable to the DS2-IS2A pair (IS2-A is referred to as IS2 hereafter). Here, we conjugated DS2-2x to an anti-rabbit antibody using a PEG16 spacer and tested its use for DNA-PAINT-ERS imaging of clathrin. Similar to that observed on DS1-2x-PEG16 and IS1, we observed rapid on-off kinetics between IS2 and the DS2-2x-PEG16 at 10–15% EC (Supplementary Movie 5). This allowed us to complete DNA-PAINT imaging and obtain well-resolved clathrin structures within 200–300 s (Fig. 3b). Owing to the high-quality single-molecule images (Supplementary Movie 5), the resulting images clearly resolved the circular clathrin-coated pits commonly seen in prior superresolution[16] and electron microscopy[17] studies. In addition, we also observed many irregularly shaped structures that may be attributed to flat clathrin plaques[17,18]. A close-up inspection of individual clathrin structures even began to reveal details reminiscent of the underlying triskelion lattice (Fig. 3b, right panels and insets). These results demonstrate the applicability of DNA-PAINT-ERS to broad DS-IS pairs, including those validated in previous DNA-PAINT experiments with slight modifications.

We next combined DS1-2x-PEG16 and DS2-2x-PEG16 in the same experiment on U2OS cells dual-labeled for microtubules (DS1) and clathrin (DS2), which was carried out in two cycles via probe exchange (exchange-PAINT[4]) in either order. Similar to the single-color experiments, we were able to obtain complete structures of the microtubules within ~200 s and those of clathrin-coated pits within ~300 s, thus completing a two-color imaging session in a little more than 8 min (Fig. 3c). There were no signs of crosstalk between the DS1-IS1 and DS2-IS2 imaging cycles, demonstrating the conserved specificity of these probes despite the use of 2x DS constructs, addition of a PEG spacer, and the presence of EC. These results suggest that multiplexed imaging with DNA-PAINT-ERS can be implemented in a manner essentially identical to those previously done with DNA-PAINT, except that now each cycle is about an order of magnitude faster.

**DNA-PAINT-ERS improves superresolution image quality.**
Besides the increased imaging speed, we also routinely obtained higher quality images using DNA-PAINT-ERS than using current DNA-PAINT. Specifically, by bringing $\tau_{on}$ from >2 s to ~0.2 s, EC allows clean, single-molecule images to be obtained even in areas of high target density, where a slow IS turnover can be problematic in causing degradations in image quality. This is demonstrated in Fig. 4a, where densely packed structures such as caveolae appeared blurred in the reconstructed images at the periphery (areas 1 and 4), although the situation improved

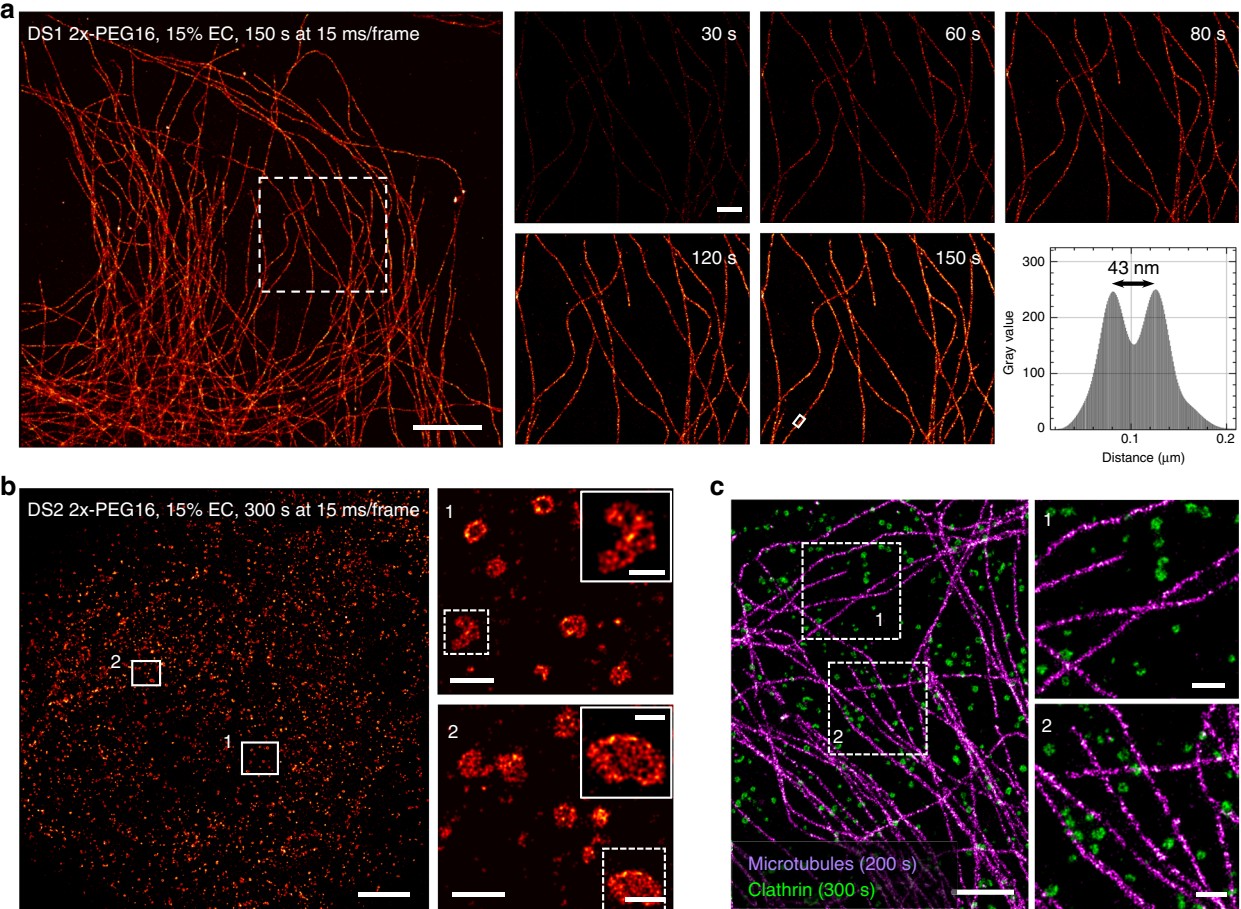

**Fig. 3 Fast superresolution imaging with DNA-PAINT-ERS using multiple DS-IS pairs. a** DNA-PAINT-ERS imaging of microtubules in U2OS cells using the DS1-2x-PEG16 construct paired with IS1-CF660R. Left panel shows the reconstructed superresolution image of the whole FOV, with zoom-in-views at different acquisition times shown on the right. Bottom right plot shows the intensity profile of the structure in the boxed area in the reconstructed image at 150 s. Similar results were obtained from at least six FOVs in two independent experiments. **b** DNA-PAINT-ERS imaging of clathrin in U2OS cells using the DS2-2x-PEG16 construct paired with IS2-CF660R. Left panel shows the reconstructed image of the whole FOV, and the right panels are the zoom-in views of the boxed areas in the image on the left. Insets in the two images on the right are the zoom-in views of the regions in the dashed boxes. Similar results were obtained from at least four FOVs in two independent experiments. **c** Two-color imaging of microtubules (purple) and clathrin (green) in U2OS cells with DNA-PAINT-ERS, using the same DS1-IS1 (tubulin) and DS2-IS2 (clathrin) constructs as used in (**a**) and (**b**). The left panel shows a ~10 × 15 μm² FOV, and the right panels are the zoom-in views of the two regions in the dashed boxes. Similar results were obtained from six FOVs in two independent experiments. Scale bars: 5 μm (**a**, left), 500 nm (**a**, right), 5 μm (**b**, left), 500 nm (**b**, right), 200 nm (**b**, right insets), 2 μm (**c**, left), and 500 nm (**c**, right).

somewhat near the center (areas 2 and 3) of the FOV. This problem is not specific to the DS2-IS2 construct as the same has been recently reported by Steher et al. on at least the 'P1' construct (from which DS1-IS1 was derived)[9]. In all these cases, the problem is likely attributed to the gradient in the rate of photobleaching from the center to the periphery of the FOV. In DNA-PAINT, photobleaching helps remove IS already bound to the sample, thus increasing the apparent unbinding rate of the IS and reducing the cluttering of the fluorescent probes to result in better resolved images. When using a Gaussian beam, the excitation power density is lower at the periphery compared with the center of the FOV, causing non-optimal localization kinetics at the periphery when the power density is optimized for best kinetics around the middle of the FOV. Stehr et al. addressed this issue by creating a flat-top illumination pattern to homogenize the power density across the FOV[9]. By contrast, inclusion of 10–15% EC in the imaging buffer drastically increases the unbinding rate of the IS to result in consistent, well-resolved structures of caveolae across the entire FOV without the negative impact of the laser intensity gradient (Fig. 4a, b).

Extending the DS-antibody spacer further helped improve the image quality. When using a short spacer (PEG4), we found that some caveolae structures detected in epi-fluorescence (visualized via Cy3 attached to the DS) failed to be reconstructed in DNA-PAINT (Fig. 4c, left column, top three rows) even with extended imaging time (Fig. 4c, left column, bottom two rows), suggesting that the absence of those structures in DNA-PAINT images was likely caused by the relative inaccessibility of the DS (DS2-1x) in those regions. The reason for this inaccessibility is currently unclear, but steric hindrance or local interactions between the DS and the antibody may be the culprit; crowding in a cellular environment may also contribute to the steric hindrance. This artifact was resolved by using PEG8, particularly with long acquisitions (20–30 min; Fig. 4c, middle column). With PEG16, all caveolae were detected and well reconstructed at 10 min, and it was no longer necessary to perform long acquisitions (Fig. 4c, right column). More examples are given in Supplementary Figs. 4–6. Aside from helping to better resolve structures, the spacers should also benefit quantitative PAINT[19] where consistent accessibility of the DS is critical. Of note, these tests were

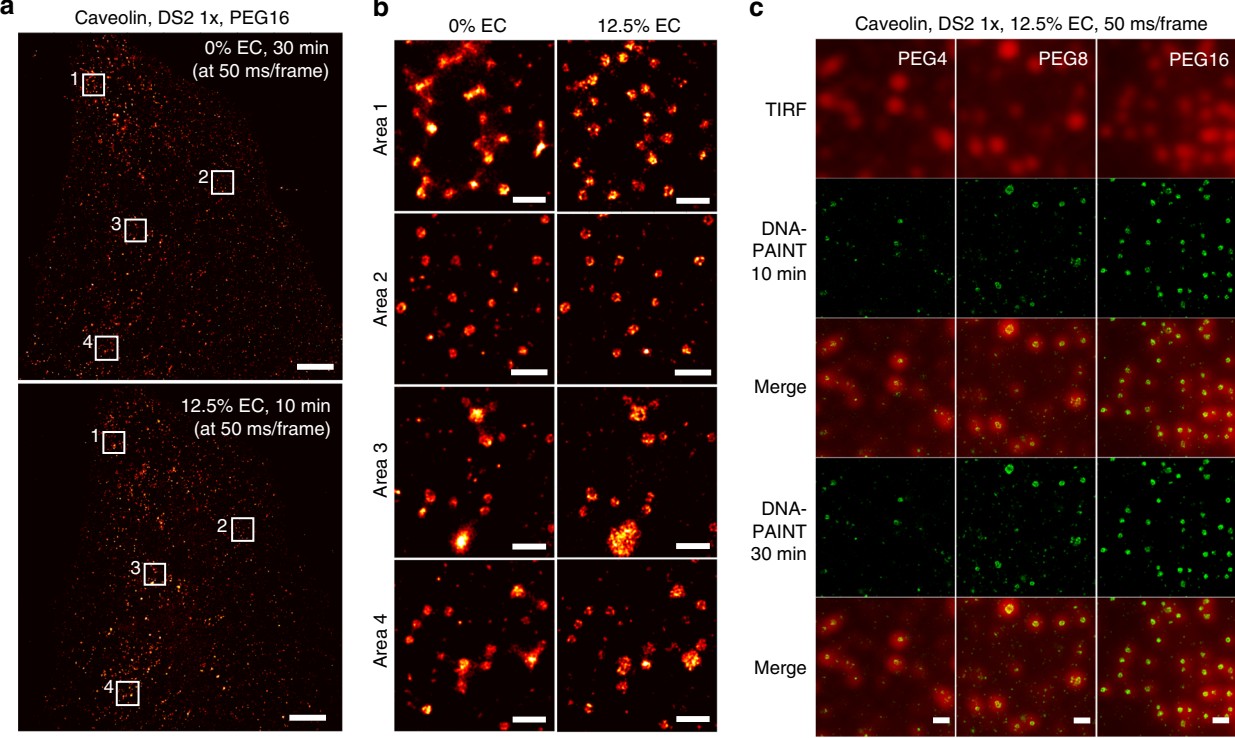

**Fig. 4 ERS improves DNA-PAINT image quality. a** Reconstructed DNA-PAINT images of caveolae in U2OS cells in the absence (top, 30 min total acquisition time) or presence (bottom, 10 min total acquisition time) of 12.5% EC using the DS2-1x-PEG16 and the IS2 pair. The comparison was repeated four times in two independent experiments. **b** Zoom-in views of the boxed areas as shown in (**a**), representing the periphery (areas 1 and 4) and the center (areas 2 and 3) of the FOV, with results in the absence of EC shown on the left and those at 12.5% EC on the right. The center areas (2 and 3) features stronger laser intensities and therefore faster photobleaching of IS2-CF660R, which helps to improve the localization kinetics (specifically the apparent 'off' rate) and image quality. Similar results were obtained from four FOVs in two independent experiments. **c** Comparing reconstructed DNA-PAINT images of caveolae in U2OS cells using DS2-1x (paired with IS2-CF660R) with a PEG4 (left), PEG8 (middle), and PEG16 (right) linker. Rows 2 and 3 comprise reconstructed images using DNA-PAINT data acquired in the first 10 min (12,000 frames at 50 ms per frame), and rows 4 and 5 comprise images using data acquired in 30 min (36,000 frames at 50 ms per frame). All imaging was performed using 2 nM IS2 in buffer C supplemented with 12.5% EC. Similar results were observed from four FOVs in two independent experiments (see Supplementary Figs. 4–6 for a subset of the example images). Scale bars: 5 μm (**a**) and 500 nm (**b** and **c**).

all carried out using the DS2-1x construct, and a PEG8 spacer might suffice if a DS-2x or -3x construct were used. In addition, although we have only tested the PEG spacers, other types of moieties may prove effective or as superior spacers.

## Discussion

In summary, we have shown that by enabling rapid DNA on-off kinetics and improved DS accessibility, DNA-PAINT-ERS allows fast superresolution imaging with high resolution and improved image quality. Compared with the recent attempt by Schueder et al.[13], DNA-PAINT-ERS not only affords similar imaging speed and resolution, but also offers important additional advantages. Most notably, DNA-PAINT-ERS does not require specific, sequence-optimized DS-IS pairs and should be applicable to most, if not all, existing DS-IS pairs (e.g., DS1 and DS2) with minor and straightforward modifications. Furthermore, DNA-PAINT-ERS uses DS-IS pairs with longer (9–11 bps) complementary sequences than current standards (8–9 bps)[4] or PS3 (6–7 bps)[13]. In fact, even DS constructs with secondary structures (e.g., DS2-2x, see Figs. 3b, c, 4; Supplementary Fig. 7; and Supplementary Movies 2 and 5) can be used. Thus, we anticipate DNA-PAINT-ERS to be compatible with a large, existing panel of DS-IS pairs to allow fast and high-quality superresolution imaging in potentially tens of colors. A systematic effort to both adopt existing[8] and identify new DS-IS pairs for DNA-PAINT-ERS is currently underway.

The excellent performance of DNA-PAINT-ERS is achieved through steps easily incorporated into current workflows without the need for special optics or an oxygen scavenger (OS). In the case of Schueder et al., an OS was used to increase photon yield from individual localization events[13]; this adds cost and complexity to the experiments, since the OS can lose activity in a matter of hours. In contrast, DNA-PAINT-ERS utilizes sequence repeats (R) and a spacer (S) that are incorporated through slight modifications to the antibody and DS preparation steps, and EC (E) is conveniently added to the standard DNA-PAINT imaging buffer (buffer C). By decoupling the localization kinetics from laser illumination, EC ensures uniform, high-quality imaging across the entire FOV on a standard wide-field imaging setup (Fig. 4), a goal previously achieved by using structured illumination[9]. By eliminating complicating factors, DNA-PAINT-ERS should facilitate automated superresolution microscopy over extended durations, such as multiplexed imaging across large FOVs[13,20] and multiple focal planes.

## Methods

**Materials**. A step-by-step guide for generating the labeling reagents, preparing immunostained samples, and performing image acquisition and initial analysis can be found at Protocol Exchange[21]. All DNA oligonucleotides were synthesized from Integrated DNA Technologies. Docking strand sequences contained a 5′ amino modifier C6 and a 3′ 6-FAM fluorophore or Cy3™ fluorophore. Imaging strand sequences contained a 3′ amino modifier that were later conjugated to CF®660R (Biotium, 92134) via succinimidyl ester chemistry. DBCO-PEG12-NHS ester was

purchased from BroadPharm (BP-24149). DBCO-PEG4-NHS ester, DBCO-Sulfo-NHS ester, and Azido-PEG4-NHS ester were purchased from Click Chemistry Tools (A134, A124, and AZ103 respectively). Invitrogen™ UltraPure™ DNase/RNase-Free Distilled water (Fisher Scientific, 10977023), sodium bicarbonate (Fisher Scientific, M-14636), sodium acetate (Sigma, 55636), and ethanol 200 proof (Fisher Scientific, 04355223) were used in oligo conjugation and purification.

AffiniPure Donkey anti-Rabbit IgG (H+L) (cat.no. 711–005–152) and AffiniPure Donkey anti-Mouse IgG (H+L) (cat.no. 715–005–150) antibodies were purchased from Jackson Immuno Research. Gibco™ Dulbecco's Phosphate-Buffered Saline (PBS) (Fisher Scientific, 14190–144), 50 kDa and 100 kDa Millipore Sigma™ Amicon™ Ultra Centrifugal Filter Units (Fisher Scientific, UFC505096 and UFC510024, respectively) were used in protein conjugation and purification. Primary antibodies used in this work include beta tubulin monoclonal antibody (ThermoFisher Scientific, 32–2600), anti-clathrin heavy chain antibody (abcam, ab21679), and anti-caveolin-1 antibody (abcam, ab2910).

The following reagents were used for immunostaining: paraformaldehyde (Sigma, P6148), Triton X-100 (Sigma, X100), 25% glutaraldehyde (Millipore Sigma, G6257), bovine serum albumin (Fisher Scientific, BP1600), sodium hydroxide (Fisher Scientific, S318-100), sodium borohydride (Sigma, 452882), Invitrogen™ Salmon Sperm DNA (Fisher Scientific, AM9680), sodium azide (Fisher Scientific, AC190381000), Gibco™ Dulbeccos PBS with calcium and magnesium (PBS+) (Fisher Scientific, 14–040–182), and 50 nm gold particles (BBI Solutions, EM. GC50/4). The fixation buffer was made from 2x PHEM buffer, which consists of 0.06 M PIPES (Sigma, P6757), 0.025 M HEPES (Fisher Scientific, BP310-500), 0.01 M EGTA (Fisher Scientific, O2783-100), and 0.008 M MgSO$_4$ (Acros, 4138–5000) in distilled water, with pH adjusted to 7 with 10 M potassium hydroxide (Sigma, 221473). The imaging buffer used in all PAINT experiments in this work is based on buffer C (500 mM sodium chloride in PBS) and contains different concentrations of EC (Fisher Scientific, AC118412500) as indicated.

**Antibody conjugation with DNA oligos.** Secondary antibodies were conjugated to docking strand (DS; see Supplementary Table 1 for the sequences) oligos via DBCO-azide click chemistry. First, DS oligos were conjugated to either DBCO-PEG12-NHS ester, DBCO-PEG4-NHS ester, or DBCO-Sulfo-NHS ester (no PEG). DBCO-ester was added in 20x molar excess to the DNA in a total reaction volume of 50 uL. The reaction ran for 3 h at room temperature and was carried out in ultra-pure water, pH adjusted to 8.5 with 1 M sodium bicarbonate. After the reaction, ethanol precipitation with 0.3 M sodium acetate at −80 °C was repeated twice on the mixture to purify the DS-DBCO product. Final DS-DBCO products were suspended in ultra-pure water. To prepare antibody-PEG4-azide conjugates, azido-PEG4-NHS was added in 100x molar excess to secondary antibodies; the reaction was carried at pH~8.5 adjusted by 1 M sodium bicarbonate and ran for 3 h at room temperature. Antibody-PEG4-azide conjugates were flowed through a 50 kDa size exclusion column and washed with PBS 15 times on the column via centrifugation 4 °C (6000 g, 2.5 min each).

Next, DS-DBCO was reacted in 5x molar excess to the antibody-PEG4-azide via copper-free click chemistry; the reaction took place overnight on a shaker at room temperature. The resulting antibody-PEGx-DS ($x = 4$, 8, or 16 depending on the PEG linker of DS-DBCO) product was purified by flowing through a 100 kDa size exclusion column and washing in PBS 5 times by centrifugation (at 6000 g, 2.5 min each, 4 °C). The final product (antibody-PEGx-DS) was suspended in PBS. Product concentrations were measured with a NanoDrop UV-Vis spectrophotometer (ThermoFisher Scientific, 2000c). Peak signals at 280 nm, 495 nm (for 6-FAM), or 550 nm (for Cy3™) were used to calculate the protein concentrations and the degrees of labeling. The antibody-PEGx-DS used in this work typically had a degree of labeling of 4–5 (i.e., 4–5 DS oligos per antibody).

**Imaging strand oligo conjugation.** Imaging strand oligos (see Supplementary Table 1 for the sequences) were reacted with 5x molar excess of CF®660R-succinimidyl ester. The reaction was carried out in ultra-pure water with pH 8.5 adjusted by 1 M sodium bicarbonate and ran on a shaker for 3 h at room temperature. The conjugated imaging strand oligos were purified by one round of ethanol precipitation using a procedure similar to that on DBCO-DS. Peak signals at 260 nm and 660 nm (CF®660 R) were used to calculate DNA concentration and degree of labeling.

**Cell culture and immunostaining.** U2OS cells (ATCC, HTB-96) were maintained in Gibco DMEM (ThermoFisher, 11995073) or phenol red-free DMEM (Fisher Scientific, 21–063–045) supplemented with 10% fetal bovine serum (Fisher Scientific, 26–140–079). U2OS cells were passaged every 3–4 days and used under passage number 15. Trypsin-EDTA (0.25%) was purchased from ThermoFisher (25200056). Corning tissue culture dishes were purchased from Fisher Scientific (08–772–22). Lab-Tek® II eight-well chambered coverglasses were purchased from ThermoFisher Scientific (155360). For superresolution imaging experiments, cells were grown on 8-well chambered coverglass in phenol red-free DMEM until 50–60% confluency on the day of fixation.

For immunostaining of clathrin or caveolin, cells were fixed for 20 min with 3.7% paraformaldehyde (PFA) in 1x PHEM buffer following a quick PBS wash.

After two PBS washes, cells were quenched with fresh 0.1% sodium borohydride in PBS for 7 min, washed with PBS (3x), and then permeabilized with 0.5% saponin in PBS for 20 min. For immunostaining of microtubules, cells were fixed for 20 min with 3.7% PFA and 0.1% glutaraldehyde (GA) in 1x PHEM before quenching with sodium borohydride and permeabilization in 0.2% Triton X-100 in PBS. Blocking in 5% BSA in PBS for 30 min was done on a rocker, followed by incubation with the primary antibody for clathrin, caveolin, or tubulin antibody (0.5 mg mL$^{-1}$ or 1:200 dilution from stock) in PBS buffer containing 3% BSA and 5% salmon sperm DNA. The incubation took place on a rocker at room temperature for 1 h. Next, cells were washed three times (5 min each) with PBS before incubation with DS-conjugated secondary antibody at a final concentration of ~8 µg mL$^{-1}$ in PBS buffer containing 3% BSA and 5% salmon sperm DNA; the incubation also took place on a rocker at room temperature for 1 h. For DS secondary antibody addition and subsequent steps, the sample was kept in the dark to avoid bleaching conjugated fluorophores. Cells were washed three times with PBS (5 min each).

For immunostaining in extracted samples (microtubules alone, or microtubules co-labeled with clathrin), cells were pre-permeabilized with cold 0.1% Triton X-100 in 1x PHEM buffer for 45 s preceding 3.7% PFA fixation in 1x PHEM for 20 min. The cells were rinsed two times with PBS and subsequently quenched with fresh 0.1% sodium borohydride in PBS for 7 min. Microtubule single staining was performed similarly to the unextracted samples described above. For co-stained samples, the cells were further permeabilized and blocked with 3% BSA and 0.2% Triton X-100 for 60 min. The cells were first labeled for beta-tubulin as described above, post-fixed with 3.7% PFA for 10 min, and then stained for clathrin as described above.

All cell samples were post-fixed for 10 min with 3.7% PFA and 0.1% GA in 1x PHEM. Before imaging, cells were incubated with 2.5% 50 nm gold particles in PBS + for 10 min, followed by a PBS wash.

**Microscopy.** All superresolution and regular fluorescence data in this work were taken on a custom single-molecule imaging system. Briefly, three lasers emitting at 488 nm (Coherent Sapphire 488, 200 mW), 561 nm (Opto Engine LLC, 150 mW), and 647 nm (Coherent OBIS 647, 140 mW) were combined and introduced into the back of a Nikon Ti-U microscope equipped with a 60× TIRF objective (Nikon, Oil immersion, NA 1.49). An $f = 400$ mm lens was placed at the back port of the microscope to focus the collimated laser light to the back aperture of the objective to achieve through-objective total internal reflection (TIR) illumination. The excitation light can be continuously tuned between epi-fluorescence and strict TIR modes by shifting the incident laser horizontally with a translational stage before entering the back port of the microscope. Most images in this work were acquired with moderately relaxed TIR, so structures such as the microtubules could be probed adequately. A custom focus stabilizing system based on detection of the reflected excitation laser was used to stabilize the focus during data acquisition.

A multi-edge polychroic mirror (Semrock, Di01-R405/488/561/635) was used to reflect the lasers into the objective and clean up fluorescence signals from the sample. Emission filters used for the 488 nm (for imaging FAM on the DS), 561 nm (for imaging Cy3 on the DS), and 647 nm (for imaging CF660R conjugated ISs) were FF01-525/45, FF01-605/64, FF01-708/75, respectively (all from Semrock). Fluorescence signals were collected through the objective by an electron-multiplied charge-coupled device (EM-CCD, Andor, iXon Ultra 897) using a typical EM gain setting at 200–300 in frame transfer mode. Unless otherwise indicated, the power density of the 647 nm laser (for DNA-PAINT imaging using CF600R conjugated IS) was typically around ~500 W cm$^{-2}$.

**Data acquisition and analysis.** Superresolution images were acquired using the open source micromanager software suite (https://micro-manager.org/)[22] and saved as OMERO TIF files. Image analyses for extracting single-molecule localization and subsequent localization filtering, sorting, and rendering was performed using in-house Matlab scripts[23]. Briefly, raw localizations were first filtered based on localization fitting parameters such as signal to noise ratio, widths of point spread functions in the x and y dimensions, aspect ratio, etc. Next, the localizations were sorted, during which events that appeared within a defined number of frames (typically 2–3) and distance (typically 80 nm) were then combined into a single event with averaged coordinates. The sorted localizations were then used for final image rendering, and the rendered images were saved as TIF files for further analysis and annotations in Fiji.

**Reporting summary.** Further information on research design is available in the Nature Research Reporting Summary linked to this article.

## Data availability
Data supporting the findings of this work are available within the paper and its Supplementary Information files. A reporting summary for this Article is available as a Supplementary Information file. The datasets generated and analyzed during the current study are available from the corresponding author upon request. The raw images have been provided in the form of curated Supplementary Figures and Movies. The source data underlying Fig. 2a–c, as well as Supplementary Fig. 2d, e are provided as a Source Data file.

## Code availability

The custom Matlab scripts wfiread [https://github.com/nanxiaolin/wfiread] and palm [https://github.com/nanxiaolin/palm] used in this work are available at GitHub.

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

## Acknowledgements

The authors thank Drs. Joe W Gray, Bruce Branchaud, Paul Spellman, Sunjong Kwon, Daniel Heineck, Yu-Jui (Roger) Chiu, and other colleagues at OHSU for their helpful discussions. Research in the Nan lab was supported by the OHSU Knight Cancer Institute, the Damon Runyon Cancer Research Foundation, the M. J. Murdock Charitable Trust, the Prospect Creek Foundation, the Cancer Systems Biology Consortium from the National Cancer Institute (CSBC, grant number U54 CA209988, PI: Joe W. Gray), and the National Institute of General Medical Sciences (grant number R01 GM132322, PI: X.N.). F.C., T.Z., M.R., J.K., S.E., and X.N. are members of and supported by the Caner Early Detection Advanced Research (CEDAR) Center of the OHSU Knight Cancer Institute.

## Author contributions

X.N. conceived and supervised the project. F.C., T.Z., and X.N. developed the initial workflow and established the use of EC and sequence repeats. J.S., K.T., and X.N. established the use of spacer and further optimized the workflow. J.S., K.T., F.C., T.Z., M.R., J.K., and X.N. acquired and analyzed the data. Y.Z., L.W., and C.P. assisted with data acquisition and analysis. J.S., K.T., and X.N. wrote the paper. F.C., T.Z., M.R., and J.K. edited the paper. S.E. offered guidance and resources for the project. All authors have reviewed and approved the paper.

## Competing interests

The authors declare no competing interests.
