## [Peer Review File · Nature Communications]

Reviewers' comments:

Reviewer #1 (Remarks to the Author):

Civitci et al. report an effort to improve the imaging speed of DNA-PAINT by the addition of ethylene carbonate (EC) into the imaging buffer. EC helped increase the unbinding speed of the imaging strand from the docking strand, and with the addition of repeating sequences and PEG spacers in the docking strand, they increased the binding kinetics. Thus they achieved DNA-PAINT with improved speeds at several minutes per image, and demonstrated two-color imaging.

The work is generally interesting, and I recommend the publication of this work in Nature Communications, with the below comments.

1. The authors should describe what their method is in the abstract. Now the name "DNA-PAINT-ERS" just comes up from nowhere.
2. What are the actual frame rates used to record the different data? Some of this information is hidden in the captions of their videos, but this should be more clearly stated in the text, captions, and the method section. For example, as they show good DNA-PAINT images in 150 s in Fig. 2a, how many frames were accumulated? If the frame rate was 67 frames/s as stated in the video caption, then 10,000 frame seems a bit low for typical SMLM? Also, in the videos shown, please add in timestamps and scale bars in each frame.
3. In figure 1B, bottom panel, for the differential images, can the authors show both positive and negative changes in signal, e.g., using different colors? Then it could show which single molecules have unbound.
4. The comparison of with and without EC in Fig. 3AB may not be very fair, since in this case it was with a docking strain optimized for performance in the presence of EC. I suggest either moving them to supplement or be more cautious in the discussion.
5. It would be useful to add in data to show whether the single-molecule brightness (photon count per unit time) is varied due to the addition of EC into the buffer. Fig. S1 alludes to this, but it would be useful to get histograms of photon counts.
6. In the introduction part, "other SMLM (t on ~ 0.1 s)⁶": 10 ms time scale may be more common for STORM/PALM.

Reviewer #2 (Remarks to the Author):

in this work, the authors propose and test a number of technical improvements for DNA-PAINT, an approach to single molecule localisation microscopy that is based on the controlled, transient hybridisation of dye-coupled DNA oligomers with their reverse-complementary counterparts that are bound to a target structure. This technique is promising as it allows for multiplexed labelling and higher resolution than other approaches such as (d) STORM and PALM.

The authors present:

a) an optimisation of the binding period for shorter, but equally specific binding events using addition of ethylene-carbonate, a reagent known from other DNA-based techniques such as FISH. The authors present very convincing, high quality data to support their claim that this speeds up DNA-PAINT imaging.

b) a multiplication of the target sequence on the DNA to increase local concentration and faster on-rate. This significantly increases signal, although the quantification is not entirely clear and more controlled quantitative data, maybe acquired in vitro should be presented.

c) the introduction of a polyethylene glycol linker between the antibody and the oligomer to make the oligomer more accessible. This furthermore increases the signal.

Overall the authors present three tweaks to DNA-PAINT that will make this technique more efficient and accessible to many and this is an important and worthwhile contribution. The experimental setup is designed nicely and the results convincing. The only problem is that the quantification for b) and c) is not clear and should be demonstrated using a clear in vitro setup rather than intrinsically uneven biological images.

I thus suggest minor revisions.

Reviewers' comments:

We thank the reviewers for their enthusiastic responses and insightful comments. Based on these comments, we have conducted additional experiments and data analyses, as summarized below in our response to specific questions and in the revised manuscript. In addition to these, we have also made minor changes throughout the manuscript to correct typos and clarify certain points (see revised text with changes highlighted). Lastly, we would like to apologize for the delays in submitting the revised manuscript due to interruptions to our research when carrying out the additional experiments. Despite the setback and time constraint, we hope that the reviewers find our responses to the below comments acceptable.

=====

Reviewer #1 (Remarks to the Author):

Civitci et al. report an effort to improve the imaging speed of DNA-PAINT by the addition of ethylene carbonate (EC) into the imaging buffer. EC helped increase the unbinding speed of the imaging strand from the docking strand, and with the addition of repeating sequences and PEG spacers in the docking strand, they increased the binding kinetics. Thus they achieved DNA-PAINT with improved speeds at several minutes per image, and demonstrated two-color imaging.

The work is generally interesting, and I recommend the publication of this work in Nature Communications, with the below comments.

1. The authors should describe what their method is in the abstract. Now the name "DNA-PAINT-ERS" just comes up from nowhere.

We completely agree. The abstract has now been updated to include more specific details of the DNA-PAINT-ERS method, so it becomes clear what our new strategies entail from reading the abstract.

2. What are the actual frame rates used to record the different data? Some of this information is hidden in the captions of their videos, but this should be more clearly stated in the text, captions, and the method section. For example, as they show good DNA-PAINT images in 150 s in Fig. 2a, how many frames were accumulated? If the frame rate was 67 frames/s as stated in the video caption, then 10,000 frame seems a bit low for typical SMLM? Also, in the videos shown, please add in timestamps and scale bars in each frame.

Thank you for noting this, and apologies for the confusion. The frame rate used in each case depends on the buffer condition, the DS/IS used, and the density of the targets. In general, we used slower frame rates (10-20 Hz) when binding and/or unbinding was slow (e.g. using 0-5% EC or DS-1x, or when imaging more sparse targets such as caveolin). For example, at 0% EC (as in previous DNA-PAINT workflows) the unbinding of IS was slow, and there would be little point acquiring data faster than 100 ms/frame. This slow frame rate was useful to better detect binding and unbinding by measuring changes between successive frames (as we did in Fig. 1B, left). By contrast, in Fig. 1D and Fig. 2 we used higher frame rates (15 – 30 ms/frame) because the buffer contained 10-15% EC and IS unbinding was much faster. We have followed your suggestion and labeled the frame rates for each dataset (both images and videos) to help keep track of the imaging conditions.

As for showing reconstructed DNA-PAINT images up to 150s in Fig. 2A, we did it for two reasons. First, at 10,000 frames (15 ms/frame or ~ 67 Hz), we were already able to collect 2 – 4 million raw localizations per $30 \times 30 \mu\text{m}^2$ area (0.6 – 1.3 million after sorting, where localizations arising from the same event were combined) to yield good reconstructions, as demonstrated in Fig. 2A. The second reason was to show that our imaging speed was comparable to that recently reported by Jungmann lab (Scheuder et al., Nature Methods, 2019, v11, p1101, where the authors showed well reconstructed DNA-PAINT images at 150s; see Supplementary Figure 13 therein). In practice, the number of frames to be acquired for each imaging session also depends on the density of the target. For most targets we have imaged with DNA-PAINT-ERS, a total acquisition of 3-6 minutes would typically suffice. This can be seen in Fig. 2 and 3. In fact, even when using DS2-1x (which is 3-4 slower than DS2-2x) we were able to acquire well resolved images of caveolae in ~ 10 minutes (Fig. 3, using 12.5% EC).

3. In figure 1B, bottom panel, for the differential images, can the authors show both positive and negative changes in signal, e.g., using different colors? Then it could show which single molecules have unbound.

Nice suggestion. We have now added the negative changes in a different color to show both the newly bound and the unbound molecules (see updated Fig. 1B; the third row in green). Of note, IS binding and unbinding are at equilibrium during DNA-PAINT, and since the binding rate was not significantly affected by EC%, the number of binding and unbinding events appeared to be similar at all EC%. Therefore, the sharp decrease in the instantaneous number of IS bound on the sample at increasing EC% translates into higher unbinding rates (or equivalently, shorter t_{on}).

4. The comparison of with and without EC in Fig. 3AB may not be very fair, since in this case it was with a docking strain optimized for performance in the presence of EC. I suggest either moving them to supplement or be more cautious in the discussion.

The DS-IS pairs used in our work (DS1-IS1, DS2-IS2, etc.) were derived from previously validated constructs (designated P1, P2, ..., in publications from the Jungmann and Yin labs). While we only made small modifications to these constructs by adding 1-2 bases to them, we agree that the new DS-IS should still be viewed as 'optimized' constructs for use with EC. That being said, the same issue in image quality as shown in Fig. 3A (top, at 0% EC) was observed by Jungmann et al. when using the 'non-optimized' constructs (Stehr et al., Nat. Comm., 2019, v10, p1268; reference #9). In other words, the image quality problem in Fig. 3 was intrinsic to DNA-PAINT and not specific to our new DS-IS constructs. Therefore, the improvements in image quality by DNA-PAINT-ERS presented in Fig. 3A-B should still hold true even when compared with images taken with the unmodified DS-IS pairs (the P1, P2, ... constructs).

The degradation in imaging quality (particularly at the periphery of the FOV) is caused by the spatial unevenness in photobleaching of the IS. Photobleaching eliminates fluorescence of those IS already bound to the sample, effectively 'unbinding' the IS. Photobleaching, however, is uneven: it is the fastest at the center of the FOV and much slower at the periphery when using a Gaussian beam. The slower photobleaching at the periphery leads to IS cluttering and reduces image quality and resolution. Stehr et al. solved this problem by using a flat-top illumination. By contrast, EC helps accelerate IS unbinding independent of photobleaching or illumination, thus yielding well-resolved images across the whole FOV, and this effect is generic to all DS-IS pairs used for DNA-PAINT.

We have followed your advice and added more discussions on this point in the main text (see page 9).

5. It would be useful to add in data to show whether the single-molecule brightness (photon count per unit time) is varied due to the addition of EC into the buffer. Fig. S1 alludes to this, but it would be useful to get histograms of photon counts.

We have added the analysis to supplementary figure 1. From this analysis we confirmed that the single-molecule brightness per unit time (frame) remained largely the same at 0-10% EC, although a 10-15% reduction in brightness was observed at 15% EC. The dimmer signal at 15% EC is likely due to the fact that a fraction of the IS stayed bound for less than a single frame. In practice, we offset this by using a slightly stronger laser illumination and/or a slightly lower EC concentration (e.g. 12.5% instead of 15%).

6. In the introduction part, “other SMLM ($t_{\text{on}} \sim 0.1 \text{ s}$)⁶”: 10 ms time scale may be more common for STORM/PALM.

Thanks for bringing this up. Yes, for STORM the t_{on} is indeed on the scale of 10 ms (varies depending on the imaging conditions such as laser power and buffer composition). For PALM, the on/off kinetics is typically somewhat slower with a t_{on} typically around 50-200 ms. We have thus changed the statement to ‘... other SMLM (t_{on} on the order of 0.01 – 0.1 s)’ to be more accurate.

=====

Reviewer #2 (Remarks to the Author):

in this work, the authors propose and test a number of technical improvements for DNA-PAINT, an approach to single molecule localisation microscopy that is based on the controlled, transient hybridisation of dye-coupled DNA oligomers with their reverse-complementary counterparts that are bound to a target structure. This technique is promising as it allows for multiplexed labelling and higher resolution than other approaches such as (d) STORM and PALM.

The authors present:

a) an optimisation of the binding period for shorter, but equally specific binding events using addition of ethylene-carbonate, a reagent known from other DNA-based techniques such as FISH. The authors present very convincing, high quality data to support their claim that this speeds up DNA-PAINT imaging.

b) a multiplication of the target sequence on the DNA to increase local concentration and faster on-rate. This significantly increases signal, although the quantification is not entirely clear and more controlled quantitative data, maybe acquired in vitro should be presented.

c) the introduction of a polyethylene glycol linker between the antibody and the oligomer to make the oligomer more accessible. This furthermore increases the signal.

Overall the authors present three tweaks to DNA-PAINT that will make this technique more efficient and accessible to many and this is an important and worthwhile contribution. The experimental setup is designed nicely and the results convincing. The only problem is that the quantification for b) and c) is not clear and should be demonstrated using a clear in vitro setup rather than intrinsically uneven biological images.

I thus suggest minor revisions.

We thank the reviewer for the positive comments and for suggesting additional investigations to better quantitate the effects of EC (E), repeating sequences (R), and spacers (S). We agree that inter- and intra-cellular heterogeneities could complicate the measurements. We had initially thought about using in vitro systems such as DNA origami for studying the effects of E/R/S but did not proceed because EC may disrupt the origami structures. We also chose to use actual cell samples in this work because, despite complexities of these samples, the results would provide a closer view of the localization kinetics when imaging real biological samples. Among other factors, the local environment of the target and the docking strand (DS) would likely have an impact on imager strand (IS) binding and localization kinetics; this impact might not be recapitulated by using in vitro systems. For example, a DS-oligo hanging off of a DNA origami may not experience the same steric hindrance as an oligo conjugated to an antibody. As a result, we may not see the same acceleration on the origami when adding a spacer to the DS.

For these reasons, we sought alternative routes to minimize the impact of cell-to-cell variations on our measurements. To determine the effect of EC, we took each E% dataset in the same field-of-view (FOV) in a sequential manner, by replacing the imaging buffer with escalating concentrations of EC (from 0% to 5%, 10%, then 15%). As such, in Fig. 1B we were comparing data from exactly the same set of targets, using images taken at 0% EC as the normalization standard, thus minimizing the impact of cell-to-cell variations. The ability to ‘re-image’ the same FOV multiple times via simple buffer exchange in DNA-PAINT worked to our advantage in this case.

Doing the same for Figs. 1D (effects of R) and 1E (effects of S) was a bit trickier, however, since it required the use of different DS-IS pairs. That was why we used signals of the DS (conjugated to a fluorophore that emits in a different channel) as a normalization standard instead. Here each pair of DNA-PAINT and epi-fluorescence images were also from the same FOV. Prompted by your question, we have come up with another experiment where we could PAINT the same FOV using the same DS and 3 different IS oligos in a sequential manner. All three IS oligos bind to the DS, but they differ in the number of binding sites on the DS and/or the spacer length, thus allowing us to directly measure the impact of repeating sequences and spacers on the localization kinetics.

The experimental design and the results are shown in Fig. R1 below, which has now also been added to our manuscript as the new Supplementary Figure 2 and Supplementary Video 2. Three imager strands, IS2-A, IS2-B, IS2-C, target the same docking strand, DS2-2x-PEG4, with different recognition sequences. IS2-A has two binding sites on the DS2-2x (Fig. R1A), with the 1st site located 2 bases (plus a PEG4 linker) away from the antibody; the second binding site is much further downstream. IS2-B has a single binding site on the DS2-2x, located 4 bps away from the antibody; IS2-C binds 2 bases further downstream on the DS, also to a single site. We note that IS2-A is the same as IS2 used in Fig. 2B and Fig. 3. As such, the three IS2 oligos have the same base compositions (4 As, 4 Ts, and 2 Gs) and should have similar melting temperatures (T_m). Indeed, using an online oligo analyzer tool (<https://www.genscript.com/tools/oligo-primer-calculation>; see also Supplementary Table 1), we found the predicted T_m s are 11.4 °C, 9.2 °C, and 11.4 °C for IS2-A, IS2-B, and IS2-C, respectively.

In this setup, IS2-B was used as a ‘baseline’ IS. Differences between IS2-A and IS2-B in their kinetics of binding to DS2 would be dominated by the difference in the number of binding sites, since IS2-A has 2 binding sites and IS2-B has only 1. Indeed, using IS2-A yielded ~3x binding events compared with using IS2-B; the difference is evident in the raw images, the difference images (showing binding), as well as in the reconstructed DNA-PAINT images (see Fig. R1A and R1B). The difference was quantitated in Fig. R1D.

Figure R1. Effect of repeating sequence and spacer length on DNA-PAINT imaging kinetics using DS2-2x. Three IS oligos, namely IS2-A, IS2-B, and IS2-C, bind to 2 sites (A, left), 1 site (B, left), and 1 site (C, left), respectively, on the same DS2-2x-PEG4. U2OS cells were fixed, permeabilized, and immuno-labeled for microtubules using a secondary antibody conjugated to the DS2-2x-PEG4 and subsequently imaged with DNA-PAINT 12.5% EC using 2 nM CF660R conjugated IS2 (-A, -B, or -C) and a frame acquisition rate of 50 ms/frame. Each field of view was imaged using all three ISes in three sequential cycles, and the sample was carefully and thoroughly washed in between the cycles without shifting the sample position. The resulting raw images (A-C, 2nd column), differential images featuring binding events (A-C, 3rd column), and resulting DNA-PAINT reconstructions (from 2,500 raw frames; A-C, 4th column) show clear distinctions in the imaging kinetics depending on the IS used. The ratios of sorted localizations between IS-2A and IS-2B or between IS-2C and IS-2B are shown in (D) and (E), respectively; here IS-2B is used as a normalization standard. Results from 4 field of views were recorded and analyzed. Scale bars, 5 μ m.

By contrast, both IS2-B and IS2-C have 1 binding site on the DS2-2x except that the binding site of IS2-C is 2 bps further away from the antibody than IS2-B. Despite this seemingly small difference, binding of IS-2C was 30-40% more efficient than IS2-C (Fig. R1B, 1C, and 1E). This result suggests that a 2-base additional space between the binding site on the DS and the affinity agent could help reduce steric hindrance for IS binding. The same spacer effect may have also favored IS2-B in its comparison with IS2-A, which may in part explain the somewhat lower enhancement by using IS2-A vs IS2-B (~3x) compared with that shown in Fig. 1C (~3.5x); in the latter case, all DS constructs had the same spacer and the same IS with the only difference being the number of sequence repeats.

Overall, these comparisons corroborate our observation that using a DS with a 2x sequence repeats yields >2x faster binding, and support our hypothesis that the >2x enhancement is likely due to the relaxed steric hindrance at the 2nd site. The results also imply that nucleotide bases between the affinity agent and the DS can also be used spacers; in fact, in DS-2x, the 1st binding site (10-12 bases) essentially acted as spacer for the 2nd. Since IS2-A is ~2x as efficient as IS2-C, a block of ~6 bases may replace PEG₈ or PEG₁₆ as an effective spacer, a scheme that we plan to test in our future work.

REVIEWERS' COMMENTS:

Reviewer #1 (Remarks to the Author):

The authors have addressed all my questions and improved the manuscript accordingly. I just have one more comment: in Supplementary figure 1, would it be possible to convert "sum intensity" into photon counts? That is typically done in SMLM, and would be more useful to readers.

Reviewer #2 (Remarks to the Author):

The authors have addressed my concerns and I suggest acceptance of the improved manuscript.

Response to Reviewer's comments

We thank the reviewers for their enthusiastic responses again. Below we address the remaining request from reviewer #1.

REVIEWERS' COMMENTS:

Reviewer #1 (Remarks to the Author):

The authors have addressed all my questions and improved the manuscript accordingly. I just have one more comment: in Supplementary figure 1, would it be possible to convert "sum intensity" into photon counts? That is typically done in SMLM, and would be more useful to readers.

Thank you. We have updated supplementary figure 1 as suggested.

Reviewer #2 (Remarks to the Author):

The authors have addressed my concerns and I suggest acceptance of the improved manuscript.